# Effect of Shape on Mesoporous Silica Nanoparticles for Oral Delivery of Indomethacin

**DOI:** 10.3390/pharmaceutics11010004

**Published:** 2018-12-23

**Authors:** Wei Zhang, Nan Zheng, Lu Chen, Luyao Xie, Mingshu Cui, Sanming Li, Lu Xu

**Affiliations:** School of Pharmacy, Shenyang Pharmaceutical University, Wenhua RD 103, Shenyang 110016, China; zhangwei9501@126.com (W.Z.); zhengnangg@hotmail.com (N.Z.); chenlu182125@163.com (L.C.); xieluyao0221@163.com (L.X.); cui_mingshu@126.com (M.C.); lisanming2018@163.com (S.L.)

**Keywords:** mesoporous silica nanoparticles, shape effect, drug delivery, indomethacin

## Abstract

The use of mesoporous silica nanoparticles (MSNs) in the field of oral drug delivery has recently attracted greater attention. However, there is still limited knowledge about how the shape of MSNs affects drug delivery capacity. In our study, we fabricated mesoporous silica nanorods (MSNRs) to study the shape effects of MSNs on oral delivery. MSNRs were characterized by transmission electron microscopy (TEM), nitrogen adsorption/desorption, Fourier transform infrared spectroscopy (FTIR), and small-angle X-ray diffraction (small-angle XRD). Indomethacin (IMC), a non-steroidal anti-inflammatory agent, was loaded into MSNRs as model drug, and the drug-loaded MSNRs resulted in an excellent dissolution-enhancing effect. The cytotoxicity and in vivo pharmacokinetic studies indicated that MSNRs can be applied as a safe and efficient candidate for the delivery of insoluble drugs. The use of MSNs with a rod-like shape, as a drug delivery carrier, will extend the pharmaceutical applications of silica materials.

## 1. Introduction

Owing to its convenience and high patient compliance, oral administration has been used widely and is the most accepted and preferred drug delivery pathway [1,2,3]. However, there are still limitations to this form of delivery, and the challenge remains to improve oral bioavailability of Biopharmaceutical Classification System (BCS) II molecules, owing to their low solubility and poor stability in the gastrointestinal tract (GI tract). In recent years, many strategies have been proposed to improve the dissolution and bioavailability of poorly water-soluble drugs, including spray drying [4,5,6], solid dispersions [7,8,9], nanosuspensions [10,11], cryogenic technologies [12,13], and prodrug approaches [14]. For example, Xu et al. prepared hydrophilic polymer-based solid dispersions by a freeze-drying technique, and enhanced the dissolution rate and oral bioavailability of valsartan [15]. Pahweenvaj Ratnatilaka Na Bhuket designed a prodrug for curcumin to improve its solubility and oral bioavailability [16]. Li et al. found that amino-modified chiral mesoporous silica nanoparticles exhibited improved drug dissolution and reduced hemolysis, which subsequently increased bioavailability [17]. It is therefore of great importance to develop various approaches to improve the water-solubility of BCS II drugs.

It has recently emerged that nanoparticle-based drug delivery systems offer advantages for orally administered drugs, owing to their ability to deliver sufficient drugs to specific sites, avoid digestion in the GI tract, and control the release and cellular uptake of incorporated drugs [18,19,20]. Of the various drug carriers explored, mesoporous silica nanoparticles (MSNs) exhibit excellent properties, such as controllable pore size, large specific surface area, and easy surface modification, as well as the ability to prevent crystallization of the encapsulated drug, which enhances its apparent solubility [21,22,23]. Therefore, scientists have synthesized MSNs with varying size, surface charge, and morphology to carry poorly water-soluble drugs and improve bioavailability [1,24,25]. Normally, MSNs were fabricated by a sol-gel method based on tetraethoxysilane, which relies on hydrolysis to provide silica source and condensate on a template, such as cetyltrimethylammonium bromide (soft surfactant template), anodized aluminum oxide (hard template), etc. Finally, a mesoporous structure is formed. For example, He et al. evaluated the feasibility of MSNs to load paclitaxel, a typical chemotherapeutic agent with poor water-solubility, and found that MSNs could be a potential carrier for poorly soluble drugs [26]. Shen et al. demonstrated that pore size and particle size of MSNs influenced the dissolution profile of ibuprofen [27]. However, there are no systematic studies about the effects of shape on absorption in vivo, especially, investigations into whether the particle shape of MSNs, controlled by alkyl alcohols, can affect the dissolution behavior and oral efficiency are still rare.

Herein, this work introduces the synthesis of MSNs with helical channels for drug delivery and reveals how the behavior of MSNs with different shapes differ in their drug delivery properties. We used a facile method to prepare MSNs with different ratios of cetyltrimethylammonium bromide (CTAB) as supramolecular template, and alkyl alcohols as co-structure-directing agents. Indomethacin (IMC), which was used as the model drug, is poorly soluble in water and has limited bioavailability when orally administered [28]. After loading IMC into MSNs, we evaluated the influence of particle shape on the efficiency of drug loading and release. Different shapes of MSNs resulted in different performances, both in vitro and in vivo. Overall, our results, highlighted here, provide a meaningful platform for the development of nanoparticle-based drug delivery systems for use in pharmaceutical science.

## 2. Materials and Methods 

### 2.1. Materials

Tetraethoxysilane (TEOS) was purchased from Aladdin (Shanghai, China). Cetyltrimethylammonium bromide (CTAB) and *n*-octanol were purchased from YuWang Chemical Reagent Corporation (Shandong, China). Deionized water was prepared by ion exchange. All other chemicals were of analytical grade and used as required without further purification. 

### 2.2. Preparation of Mesoporous Silica Nanoparticles (MSNs)

Rod-like mesoporous silica nanoparticles (MSNRs) were fabricated by a previously described method [29], as shown in Scheme 1. First, 0.2 g CTAB was dissolved in 100 mL deionized water in an 80 °C water bath, and 0.086 mL *n*-octanol was added dropwise into the solution at a stirring rate of 600 rpm. After 30 min, 0.70 mL NaOH aqueous solution (2.0 mol/L) was added to the above mixture, and 1.0 mL TEOS was added dropwise into the mixture under stirring. Stirring was continued for 2 h for further silica condensation. The resultant white precipitate was filtered and dried. To remove the template CTAB, the particles were boiled in a mixture of 40 mL ethanol and 5.0 mL of 36.0 wt % aqueous HCl for 10–20 min. Finally, the dried sample was calcined at 550 °C for 6 h.

The mesoporous silica nanospheres (MSNSs) were prepared by using a previously reported method [30]. Briefly, 0.28 g NaOH and 1 g CTAB were dissolved in 480 mL deionized water in an 80 °C water bath. Subsequently, 5 mL TEOS was added dropwise into the solution and stirred for 2 h. The mixture was centrifuged and washed with ethanol three times, and the particles were resuspended in ethanol. CTAB was removed by using the same method described for MSNRs.

### 2.3. Drug Loading

The drug-loading process consisted of a combination of adsorption equilibrium and solvent evaporation [20]. Briefly, 50 mg IMC was dissolved in 1 mL acetone to obtain a drug solution, mixed with 150 mg MSNs, and stirred for 24 h. The solvent was removed completely by using vacuum drying. To determine the drug-loading capacity, the loaded IMC was extracted completely by using methanol with ultrasound from an accurately weighed quantity of IMC-loaded MSNs, and the IMC content was measured by using ultraviolet spectroscopy (UV-1750, Shimadzu, Kyoto, Japan) at 320 nm [21]. Drug-loading capacity (DLC) of IMC-loaded MSNs was calculated from the following equation:(1)$$\mathrm{DLC}\;\%=\frac{m_{DiN}}{m_{D}+m_{N}}\times 100\%,$$
where $m_{DiN}\;\left(mg\right)$ is the amount of drugs in nanoparticles, and $m_{D}\;\left(mg\right)$ and $m_{N}\;\left(mg\right)$ are initial weights of drug and nanoparticles in the system, respectively.

### 2.4. Characterization

#### 2.4.1. Fourier Transform Infrared Spectroscopy (FTIR)

Samples were milled to obtain uniform powder and mixed with dried KBr, then transparent and thin KBr disks were prepared on a hydraulic press. Record the FTIR spectra (Spectrum 1000, Perkin Elmer, Waltham, MA, USA) of samples ranging from 400 to 4000 cm^−1^ in transmittance mode, and the resolution is 1 cm^−1^. 

#### 2.4.2. Transmission Electron Microscopy (TEM)

Tecnai G2 20 TEM instrument (FEI, Hillsboro, OR, USA), which was operated at 200 kV, was used to characterize structures of MSNRs and MSNSs. First of all, both two samples were ultrasonically dispersed in ethanol and then dropped on carbon-coated copper grids. Finally, dry at 25 °C for 12 h and observe under electron microscopy.

#### 2.4.3. Small-Angle X-ray Diffraction (Small-Angle XRD)

An X-ray diffractometer, which generated X-rays at 30 mA and 30 kV by using a Ni-filtered CuKa line as radiation source, was used to obtain small-angle XRD patterns of samples. The diffraction angle changed from 1° to 6°.

#### 2.4.4. Nitrogen Adsorption/Desorption Measurement

In order to study the pore structure, we applied a SA3100 surface area and pore size analyzer (Beckman Coulter, Brea, CA, USA) to obtain the nitrogen adsorption/desorption isotherms. The specific surface area (*S*_BET_) and the pore size distributions (PSDs) were evaluated by using the Brunauer–Emmett–Teller (BET) method and Barrett–Joyner–Halenda (BJH) method, respectively.

#### 2.4.5. XRD

XRD patterns of samples were collected by using an X-ray diffractometer (X’Pert PRO, PANalytical B.V., Almelo, The Netherlands) with a CuKa line, which generates X-ray at 30 mA and 30 kV, in order to detect whether crystalline phase exists. Data were obtained when X axis (2θ) changed from 0° to 80°.

### 2.5. In Vitro Release

ZRD6-B dissolution tester (Shanghai Huanghai Medicament Test Instrument Factory, Shanghai, China) was used to evaluate the ability of in vitro dissolution with a USP paddle method (50 rpm, 37 °C). Samples of IMC, IMC-MSNRs, and IMC-MSNSs (containing 10 mg IMC) were exposed to enzyme-free simulated intestinal fluid (pH 6.8). At certain time intervals, a 5 mL aliquot of the dissolution medium was removed, followed by filtering and analyzing at 320 nm by using a UV-1750 spectrophotometer (Shimadzu, Kyoto, Japan), and an equivalent amount of fresh medium was immediately added to maintain a constant dissolution volume. All measurements were repeated for three times.

### 2.6. Cell Culture and Cytotoxicity

Caco-2 cells were cultured in Dulbecco’s modified Eagle’s medium (DMEM; Gibco, Carlsbad, CA, USA) supplemented with 10% (*v*/*v*) fetal bovine serum (FBS; HyClone), 1% (*v*/*v*) l-glutamine, 1% penicillin, 1% (*v*/*v*) nonessential amino acids, and streptomycin (100 IU/mL) at 37 °C in 5% CO_2_. The in vitro cytotoxicity of MSN samples were measured by using the MTT (3-(4,5-dimethylthiazol-2-yl)-2,5-diphenyltetrazolium bromide) assay. Briefly, the cells were seeded into 96-well plates at a density of 1 × 10^4^ cells/well. Different concentrations of MSNs (0.01, 0.02, 0.05, 0.1, 0.2, 0.5, and 1 mg/mL) were prepared and added to the cells for 24 h; subsequently, 20 μL MTT solution (5 mg/mL) was added into each well, and the plates were incubated for a further 4 h. The medium was then removed and DMSO (150 μL) was added to dissolve the MTT formazan crystals. Finally, the absorbance of the formazan product at 490 nm was measured by using a ThermoFisher Microplate Reader [30].

### 2.7. In Vivo Pharmacokinetic Study

All procedures in the in vivo pharmacokinetic study were carried out according to the Guidelines for Animal Experimentation of Shenyang Pharmaceutical University (Shenyang, China) and with the approval of the Animal Ethics Committee of the institute. Male Sprague-Dawley rats (body weight 200 ± 20 g) were randomly divided into three groups (*n* = 3 for each studied group). Prior to the experiments, the rats were fasted overnight with free access to water. Aqueous suspensions of IMC-MSNRs, IMC-MSNSs, or IMC at 40 mg/kg were orally administered, respectively, and blood samples (0.5 mL) were collected at predetermined time points (0.5, 1, 2, 3, 4, 6, 8, 12, 24, and 32 h) in microcentrifuge tubes containing heparin by retro-orbital venipuncture technique. The blood samples were immediately centrifuged (10 min, 5000× *g*) and the supernatant was collected for the HPLC analysis of IMC. 

Plasma samples were processed as follows: each plasma sample (200 μL) was mixed with 20 μL of an internal standard solution (0.5 mg/mL naproxen), 90 μL 10% K_2_HPO_4_, and 1 mL dichloromethane, and then vortexed for 3 min. After centrifugation at 10,000 rpm (6900× *g*) for 4 min, the upper water layer was removed, and the organic layer was then evaporated in a gentle stream of nitrogen at room temperature. The residue was resuspended in 100 μL of the mobile phase. After vortex mixing and centrifugation, a 20 μL aliquot was analyzed by using HPLC.

### 2.8. Statistical Analysis

All the data were presented as Mean ± SD. Unpaired Student’s t-test was used to do statistical analysis. Statistical significance was set as * *p* < 0.05, ** *p* < 0.01.

## 3. Results

### 3.1. Synthesis and Characterization of MSNs

#### 3.1.1. Synthesis and Morphology of MSNs 

For the synthesis of MSNs, CTAB and alkyl alcohol were adopted as the template and co-structure-directing agent, respectively. From the TEM images in Figure 1, it was clear that the MSNRs had a rod-like shape and the other MSNs were spherical. From the comparison of the two synthesis processes, we determined that the alkyl alcohol was important for controlling the morphologies of the MSNRs. It has been reported that the aspect ratio (AR) of nanoparticles increases as the alkyl chain length of the alcohols increases. It has also been proven that alcohols have the capacity to decrease the critical micelle concentration of CTAB in aqueous solution, triggering the formation of rod-like or worm-like micelles from spherical micelles [31]. In addition, the MSNRs showed definite lattice fringes, which indicated a helical pore architecture. In this phenomenon, it has been reported that the lengths of the particles increase as the alkyl chain lengths of the alcohol increases. It has also been proposed that the reduction of surface free energy, owing to the hemispherical structure present at the terminal of rod-like silica, was responsible for the formation of the helical structures [31,32]. Therefore, by applying the alkyl alcohols, the synthesis of nanoparticles with helical structure and rod shape is feasible. 

#### 3.1.2. Small-Angle XRD

The small-angle XRD patterns of MSNs are shown in Figure 2a. MSNRs presented a maximal peak at approximately 2.4°–2.5° 2θ, indicating the formation of the mesostructure, which was in good agreement with the TEM images (Figure 1). For MSNSs, a broader peak was identified at approximately 2.2°–2.3° 2θ, demonstrating that this mesostructure was less well-ordered than that of MSNRs [33,34].

#### 3.1.3. Nitrogen Adsorption/Desorption

Nitrogen adsorption/desorption isotherms and pore size distribution curves of MSNRs are presented in Figure 2b,c, respectively, and the calculated parameters are displayed in Table 1. The nitrogen adsorption/desorption isotherms of MSNs were typical type IV isotherms in accordance with the IUPAC classification, which indicated the mesoporous structures [12]. The pore size distribution curves showed that the pore diameters of MSNRs and MSNSs were 5.8 and 4.7 nm, respectively.

### 3.2. Drug Loading

DLC of MSNRs and MSNSs was 29.04% and 22.29%, which were calculated by Equation (1). The effective inclusion of IMC into MSNRs and MSNSs was verified by FTIR analysis. As displayed in Figure 3a, the FTIR spectrum of pure IMC showed its characteristic peaks at 1716 and 1691 cm^−1^, which were assigned to the carbonyl groups of the carboxylic acid and amide, respectively [35]. However, after incorporation into MSNRs and MSNSs, both spectra displayed a marked decrease in carbonyl stretching peaks. In addition, a slight shift to the lower wavenumbers, of 1705 cm^−1^ and 1681 cm^−1^ in MSNRs, and 1698 cm^−1^ and 1676 cm^−1^ in MSNSs, occurred, which corresponded to the acid and amide groups, respectively. It has been proposed that the silanol groups on the MSNs surface, which provide active sites for interaction with drug molecules, are beneficial for achieving high drug-loading content [36,37]. We attributed this phenomenon to the formation of hydrogen bonds between IMC molecules and silanol groups.

XRD analysis was adopted to detect the presence of a crystalline phase, which was also expected to provide indirect evidence of the effective entrapment of IMC into MSNs. As shown in Figure 3b, the absence of diffraction peaks showed that MSNRs and MSNSs were amorphous single-phase materials. The diffraction pattern of IMC showed intense and characteristic diffraction peaks, which were indicative of the highly crystalline characteristics of IMC (Figure 3b, IMC). However, after incorporation into MSNRs, no crystalline IMC was detected in the XRD pattern of IMC-MSNs (Figure 3b, IMC-MSNRs, and IMC-MSNSs), which suggested that the IMC was loaded into both MSNs carriers in an amorphous state [38,39]. It was supposed that the crystallization of IMC was prevented when it was loaded into the narrow pores and mesoscopic channels of MSNs, owing to the space confinement, which rendered the IMC into a disordered amorphous state [35].

### 3.3. In Vitro Release 

The dissolution profiles of IMC, IMC-MSNRs, and IMC-MSNSs are depicted in Figure 4. It was assumed that the release behavior of the drug loaded in mesoporous silica was mainly a diffusion-controlled process, due to the mesostructure [40]. As shown in Figure 4, the cumulative release of the IMC encapsulated in MSNRs reached up to 100% within 1 h, whereas release of the bulk drug only reached 27% after 1.5 h. Regardless of the pore structure and silica shape, the two silica samples both conferred an extreme dissolution-improving effect on IMC, which was supposed to result from the conversion of the crystalline state of IMC to an amorphous phase, which could further dramatically enhance the apparent solubility of poorly water-soluble drugs owing to its higher energy state [29]. In addition, the increased dissolution area of IMC after loading into the mesoporous silica, and the hydrophilicity of the silica surface, were also deemed to be important for enhancing the dissolution of IMC.

In addition, it was found that the dissolution of IMC from MSNRs was faster than that from MSNSs, which indicated that the different dissolution behavior may be attributed to different pore architectures of MSNs. MSNRs possessed relatively more ordered helical channels and larger pore diameters than MSNSs, thus allowing a faster dissolution rate of loaded IMC. Above all, the in vitro release test demonstrated that the release behavior of drug-loaded MSNRs can mainly be ascribed to the particle type, pore diameter, and the order level of helical channels [41].

In order to study the release kinetic of IMC, we applied DDSolver software to fit into several kinetic models, including zero-order, first-order, quadratic, Higuchi, Hixson–Crowell, Weibull, Baker–Lonsdale, Peppas–Sahlin, and Korsmeyer–Peppas [42,43,44,45]. As is shown in Table 2, the release profile of IMC and IMC-MSNSs are fit to first-order equation ($F=53.971\times \left(1-e^{-0.493t}\right);r^{2}=0.9910$; $F=98.587\times \left(1-e^{-2.255t}\right);r^{2}=0.9980$ ) and that of IMC-MSNRs is acceptable by Higuchi equation ($F=114.014\times t^{0.5};r^{2}=0.9968$), which are better than other kinetic models. The diffusion process described by the Higuchi equation conforms to the Fick rate, and the release of poorly water-soluble drugs is mainly through the diffusion of many curved channels, which corresponds to our conclusion that the release behavior is probably related to the order level of helical channels.

### 3.4. Cytotoxicity

The cytotoxicity of MSNs to Caco-2 cells was evaluated by using the MTT assay. As shown in Figure 5, the cytotoxicity of MSNs was considered negligible because the cell viability of Caco-2 cells incubated with MSNs for 24 h was approximately 100%. This result demonstrated that MSNs were not cytotoxic and could be applied as safe drug delivery systems [46].

### 3.5. In Vivo Pharmacokinetic Study

The bioavailability study was conducted to evaluate the ability of IMC-loaded MSNRs to enhance drug absorption. The various pharmacokinetic parameters are shown in Table 3. The mean plasma concentration–time profiles of the different formulations following oral administration are depicted in Figure 6. Notably, the bioavailability of IMC-MSNRs was approximately 4.0-fold and 2.2-fold higher than IMC solution and IMC-MSNSs, which achieved the highest *C*_max_. Similarly, the area under the plasma concentration–time curve (AUC) for IMC-MSNRs was significantly different than that for IMC-MSNSs and IMC solution, and was approximately 1.3-fold and 2.2-fold higher, respectively. As reported, nanorods have a longer blood circulation in the body and more easily overcome rapid clearance by the reticuloendothelial system (RES) in comparison to nanospheres. It is of note that the dissolution of IMC from MSNRs was faster than that from MSNSs, which may have resulted from the helical pore structure and larger surface area-to-volume ratio of the MSNRs. The surface area-to-volume ratio of MSNRs was approximately 1.3-fold higher than that of MSNSs, which could therefore lead to a faster dissolution rate of the loaded IMC. Therefore, the dissolution-enhanced ability of MSNRs can improve IMC oral absorption; therefore, MSNRs may be candidates as drug delivery systems for drugs currently limited by low bioavailability. 

## 4. Discussion

With the properties of convenience and patient compliance, oral administration has been adopted as the most accepted and preferred drug delivery pathway. However, there are several complicated physical barriers, such as gastric acid, enzymes, mucus, and intestinal epithelial cells. Consequently, the design of functional carriers to overcome these multiple barriers is critical. Over the past decades, mesoporous silica nanoparticles have shown enhanced delivery efficiency and increased oral bioavailability for poorly water-soluble drugs, and its size, shape, surface charge, and morphology have also been extensively researched. Mesoporous silica nanoparticles have a longer residence time in GI tract, and nanorods have been confirmed with superiority with regard to the issue of diffusion into the intestine. Yu et al. noted that, compared to mesoporous silica nanospheres, rod-like particles are not easily trapped by mesh structure of mucus and diffuse faster because MSNRs can rotate and jump in mucus [24]. In a previous study, we also found that the cellular uptake efficiency and internalization mechanism were both dependent on the shape, in which the caveolae-dependent pathway was involved in the uptake of MSNRs and that clathrin-dependent endocytosis contributed to the behavior of MSNSs, respectively [30]. In general, the caveolae-dependent pathway can translocate part of nanoparticles to endoplasmic reticulum or Golgi, leading to a higher uptake efficiency, while MSNSs pass through the clathrin-dependent pathway, and most particles are transferred into endosome and recycled outside the cell [47,48]. That is why MSNRs show higher endocytosis efficiency. All results have provided critical insights into the idea that rod-shaped particles offer great potential to improving the bioavailability of water-soluble drugs. Furthermore, we assumed that they would also suitable for poorly soluble drugs, and thereby extended the application scope of advantageous shapes. 

In this work, we found that shape could also be controlled by the addition of small molecules, such as *n*-octanol, and that it was easy to synthesize MSNRs and MSNSs. IMC, a poorly water-soluble drug, was used as a model drug to study the effect of particle shape. Both MSNRs and MSNSs led to significant improvements in plasma concentration in comparison with IMC solution; however, MSNRs exhibited the best in vivo pharmacokinetics, indicating considerable enhancement of oral bioavailability. Unexpectedly, we found that particles with helical pore structure were also obtained easily by the addition of n-octanol, which was consistent with previously reported conclusions [29,31]. We assumed it was the most likely explanation for the improvement in drug release in vitro compared with MSNSs and believed that the exact reason should be further studied. Given these properties, it appears that MSNRs may be more effective for the oral delivery of poorly water-soluble drugs than the nanospheres that are currently used.

## 5. Conclusions

In this study, MSNRs were prepared by a facile method using CTAB as template through the introduction of an achiral alcohol as a co-structure-directing agent. The use of TEM, FTIR, nitrogen adsorption/desorption, and XRD confirmed the synthesis of two types of MSNs and the effective incorporation of IMC into the mesopores. It was of interest that the MSNRs conferred an excellent dissolution-enhancing effect and led to better oral bioavailability of IMC; the reason was attributed to the ordered helical channels and larger surface area-to-volume ratio of MSNRs. In addition, the in vitro cytotoxicity showed the negligible influence of MSNRs on the viability of Caco-2 cells. Given the facile synthesis process and excellent drug delivery capacity, mesoporous silica with helical channels and a rod-like shape can be considered a good candidate for drug delivery.

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
