# Peer review of "Effect of Shape on Mesoporous Silica Nanoparticles for Oral Delivery of Indomethacin"

_pharmaceutics, 2018, doi:10.3390/pharmaceutics11010004_

Round 1

Reviewer 1 Report

The work is interesting. However statistics is completely missed therefore any conclusion is exaggerated. My suggestion is major revision and  resubmission after statistical treatment of the data.

Author Response

Response to Reviewer 1 Comments

Thank you very much for your review, the main corrections in the paper and the responds to the reviewer’s comments are as follows:

Point 1: The work is interesting. However, statistics is completely missed therefore any conclusion is exaggerated. My suggestion is major revision and resubmission after statistical treatment of the data.

Response 1: We are very sorry for our negligence of data statistics. According to your suggestion, we have added the statistical results after calculation. As is shown in Fig.4 and Fig.6, statistical significance analysis has been included.

Fig. 4. Release profile of IMC-MSNRs, IMC-MSNSs and IMC. Statistical significance is represented by *p< 0.05, **p< 0.01, ***p < 0.001.

Fig. 6. Plasma concentration vs time profile of different IMC formulations after oral administration (Mean ± SD, n=3). Statistical significance is represented by *p< 0.05, **p< 0.01, ***p < 0.001.

Reviewer 2 Report

Major revision:

Please read the paper carefully for English language style, grammar and spelling, and make appropriate corrections and changes.

The graphical abstract is too complex and repeats Figs from the manuscript. It should summarize the contents of the article in a concise, pictorial form designed to capture the attention of a wide readership online. A simple cartoon capturing the essence of the work is preferred.

The authors state that these nanovectors are useful for oral delivery of insoluble drugs. However, all this study is focused only in one compound, Indomethacin. In my opinion, if the authors are searching for a novel platform for drug delivery, this work should include, at least, two different molecules in terms of pK, logP, permeability, BCS system, etc. 

As the authors explained, MSNs are new nanoparticles. So, it is almost sure that the release of any encapsulated molecule inside this system will be highly modified. However, no data about the Kinetic models for the release of Indomethacin is presented. It is a very crucial data when a novel delivery system is developed.

it is important to discuss and explain why the shape of MSNs so influences the bioavailability of indomethacin. As a consequence the results should be detailed and discussed. 

In the introduction add more information about MSNs production methods, advantages and disadvantages

This document needs a section with statistics. Statistical analysis should be performed for the different methods, namely in vitro cytotoxicity studies and in vivo studies.

Minor revision:

Line 66 Cetyltrimethylammoniumbromide(CTAB) and  n-octanol were purchased

Line 72 - n-octyl alcohol or n-octanol (needs to be standardized)

In Section 2.3 Drug loading – insert the equation to determine the drug loading.

Add in the in vitro citotoxicity assay the positive and negative control.

Table 2 - (mean±SD, n=x)

Fig. 6 - (mean±SD, n=x)

In vitro/ in vivo in italic

Author Response

Response to Reviewer 2 Comments

Thank you very much for your review, the main corrections in the paper and the responds to the reviewer’s comments are as follows:

Point 1: Please read the paper carefully for English language style, grammar and spelling, and make appropriate corrections and changes.

Response 1: We have modified the manuscript and we also employed an English-language editing service, Editage, to polish our manuscript. The certification has been attached.

Point 2: The graphical abstract is too complex and repeats Figs from the manuscript. It should summarize the contents of the article in a concise, pictorial form designed to capture the attention of a wide readership online. A simple cartoon capturing the essence of the work is preferred.

Response 2: Thank you for your recommendation, we have re-drawn the graphical abstract to make it simple and clear.

Graphical abstract

Point 3: The authors state that these nanovectors are useful for oral delivery of insoluble drugs. However, all this study is focused only in one compound, Indomethacin. In my opinion, if the authors are searching for a novel platform for drug delivery, this work should include, at least, two different molecules in terms of pK, logP, permeability, BCS system, etc.

Response 3: It is really true as you suggested that at least two different molecules should be included if a novel platform to deliver drug is established. In fact, experiments related to the release kinetics and bioavailability of other poor water-soluble drugs, such as Carvedilol, Doxorubicin, etc. are still in progress. We hope that we can explore the inner connection and mechanism between the properties of drug molecules and oral bioavailability as expected.

Point 4: As the authors explained, MSNs are new nanoparticles. So, it is almost sure that the release of any encapsulated molecule inside this system will be highly modified. However, no data about the Kinetic models for the release of Indomethacin is presented. It is a very crucial data when a novel delivery system is developed.

Response 4: According to the opinion, we applied DDSolver software to fit into several kinetic model, including zero-order, first-order, Quadratic, Higuchi, Hixson-Crowell, Weibull, Baker-Lonsdale, Peppas-Sahlin and Korsmeyer-Peppas. As is shown in Lines 248-261 and Table 2 of Page 7-8, the realease profile of IMC and IMC-MSNSs are fit to first-order equation (;  ) and IMC-MSNRs is acceptable by Higuchi equation (), which are better than other kinetic models. The diffusion process described by the Higuchi equation conforms to the Fick rate and the release of poorly water-soluble drugs is mainly through the diffusion of many curved channels, which is corresponding to our conclusion that the release behavior is probably related to the order level of helical channels.

Table 2. Release rate constants and r2 coefficients obtained from release data fitting analyses based on several kinetic equations.

IMC

IMC-MSNSs

IMC-MSNRs

Zero-order         
   F = C + kt

F = 0.845 + 19.620t         
   r2 = 0.9704

F = 21.333 + 61.072t
      r2=0.7858

F = 20.326  + 111.671t   
  r2=0.8883

First-order             
   F = a ×(1-e-kt)

F = 53.971 ×(1-e-0.493t)  
   r2=0.9910

F = 98.587 ×(1-e-2.255t
  r2=0.9980

F = 107.487 ×(1-e-3.127t
  r2=0.9859

Quadratic            
    F = 100 × ( k1×t2 +   k2×t )

F = 100 × (-0.050t2   + 0.262t ) 
  r2=0.9923

F = 100 × (-0.713×t2   + 1.676t )
    r2=0.9824

F = 100 × ( -1.732×t2   + 2.623t ) 
  r2=0.9602

Higuchi           
  F=k×t0.5

F=19.235×t0.5            

r2=0.8765

F=85.081×t0.5       
   r2=0.9566

F=114.014×t0.5       
   r2=0.9968

Hixson-Crowell         
  F =100 × [1-(1-kt)3]

F =100 × [1-(1-0.075t)3
  r2=0.9823

F =100 × [1-(1-0.619t)3
  r2=0.9896

F =100 × [1-(1-0.980t)3
  r2=0.9794

Weibull               
  F = 100×{1-e[-(t^β)/α]}

F = 100×{1-e[-(t^0.935)/4.282]
  r2=0.9872

F = 100×{1-e[-(t^1.025)/0.446]
  r2=0.9979

F = 100×{1-e[-(t^0.991)/0.277]
  r2=0.9810

Baker-Lonsdale
  1.5×[1-(1-F/10)2/3]-F/100=kt

1.5×[1-(1-F/10)2/3]-F/100=0.007t
   r2=0.8642

1.5×[1-(1-F/10)2/3]-F/100=0.198t 
  r2=0.9450

1.5×[1-(1-F/10)2/3]-F/100=0.370t 
  r2=0.9627

Peppas-Sahlin           
   F = k1 × t0.5 +   k2t

 F = 4.800× t0.5 + 15.693t  r2=0.9783

 F = 93.402 × t0.5 - 9.030t
     r2=0.9539

 F = 113.394 × t0.5 + 0.849t  r2=0.9963

Korsmeyer-Peppas       
  F = k × tn

 F = 20.608 × t0.848      
  r2=0.9828

 F = 84.808 × t 0.488        
  r2=0.9507

 F = 114.483 × t0.506    
  r2=0.9963

Point 5: It is important to discuss and explain why the shape of MSNs so influences the bioavailability of indomethacin. As a consequence, the results should be detailed and discussed.

Response 5: To be clearer and in accordance with the reviewer concerns, we have discussed and explained why the shape of MSNs influences the bioavailability of indomethacin in Lines 277-278,293-313, Page 9-10.

Point 6: In the introduction add more information about MSNs production methods, advantages and disadvantages.

Response 6: According to the Reviewer’s suggestion, we have added relative information about MSNs production methods, advantages and disadvantages in Lines 39-56, Page 2.

Point 7: This document needs a section with statistics. Statistical analysis should be performed for the different methods, namely in vitro cytotoxicity studies and in vivo studies.

Response 7: We are very sorry for our negligence of data statistics. According to the comments, in in vivo profile and in vitro study, we have provided statistical significance analysis. For Fig.5, we intended to study the toxicity of the naked mesoporous silica nanoparticles (MSNSs and MSNRs) to Caco-2 cells to prove the safety of carriers. The results indicated that the ability of each group to kill cells was very weak and almost equal, so there was no significant differences among the groups.

Point 8: Line 66 Cetyltrimethylammoniumbromide (CTAB) and n-octanol were purchased

Response 8: We have changed “was” to “were” according to the opinion. (Line 72, Page 3)

Point 9: Line 72 - n-octyl alcohol or n-octanol (needs to be standardized)

Response 9: We standardized the name as n-octanol throughout the text according to the comment.

Point 10: In Section 2.3 Drug loading – insert the equation to determine the drug loading.

Response 10: We inserted the equation according to the comment (Lines 96-99, Page 3). “Drug loading capacity (DLC) of IMC-loaded MSNs was calculated from the following equation:

                                  (1)

where  is the amount of drugs in the nanoparticles,  and are initial weights of drug and nanoparticles in the system, respectively.

Point 11: Add in the in vitro citotoxicity assay the positive and negative control.

Response 11: In the in vitro cytotoxicity assay, we intend to study the toxicity to Caco-2 cells of the naked mesoporous silica nanoparticles at the range of 0.01-1 mg/mL. So, we chose Dulbecco’s modified Eagle’s medium (DMEM) without serum as control and we did not set positive control. We have already added the control group in the graph.

Point 12: Table 2 - (mean±SD, n=x)

Response 12: We have added “mean±SD, n=3” in the manuscript.

Point 13: Fig. 6 - (mean±SD, n=x)

Response 13: We have added “mean±SD, n=3” according to the suggestion.

Point 14: In vitro/ in vivo in italic

Response 14: We have changed “in vivo/ in vitro” to “in vivo/ in vitro” throughout the manuscript according to the comment.

Round 2

Reviewer 1 Report

The authors answered my comments. The revised version is improved. My suggestion is acceptance in the present form

Reviewer 2 Report

Accept in present form